# Polyphenols Modulating Effects of PD-L1/PD-1 Checkpoint and EMT-Mediated PD-L1 Overexpression in Breast Cancer

**DOI:** 10.3390/nu13051718

**Published:** 2021-05-19

**Authors:** Samia S. Messeha, Najla O. Zarmouh, Karam F. A. Soliman

**Affiliations:** 1Division of Pharmaceutical Sciences, College of Pharmacy & Pharmaceutical Sciences, Institute of Public Health Florida A&M University, Tallahassee, FL 32307, USA; samia.messeha@famu.edu; 2Faculty of Medical Technology-Misrata, Libyan National Board for Technical & Vocational Education, Misrata LY72, Libya; najlazar@yahoo.com

**Keywords:** programmed death-ligand 1, Epithelial-to-Mesenchymal Transition, polyphenols, triple-negative breast cancer, breast cancer

## Abstract

Investigating dietary polyphenolic compounds as antitumor agents are rising due to the growing evidence of the close association between immunity and cancer. Cancer cells elude immune surveillance for enhancing their progression and metastasis utilizing various mechanisms. These mechanisms include the upregulation of programmed death-ligand 1 (PD-L1) expression and Epithelial-to-Mesenchymal Transition (EMT) cell phenotype activation. In addition to its role in stimulating normal embryonic development, EMT has been identified as a critical driver in various aspects of cancer pathology, including carcinogenesis, metastasis, and drug resistance. Furthermore, EMT conversion to another phenotype, Mesenchymal-to-Epithelial Transition (MET), is crucial in developing cancer metastasis. A central mechanism in the upregulation of PD-L1 expression in various cancer types is EMT signaling activation. In breast cancer (BC) cells, the upregulated level of PD-L1 has become a critical target in cancer therapy. Various signal transduction pathways are involved in EMT-mediated PD-L1 checkpoint overexpression. Three main groups are considered potential targets in EMT development; the effectors (E-cadherin and Vimentin), the regulators (Zeb, Twist, and Snail), and the inducers that include members of the transforming growth factor-beta (TGF-β). Meanwhile, the correlation between consuming flavonoid-rich food and the lower risk of cancers has been demonstrated. In BC, polyphenols were found to downregulate PD-L1 expression. This review highlights the effects of polyphenols on the EMT process by inhibiting mesenchymal proteins and upregulating the epithelial phenotype. This multifunctional mechanism could hold promises in the prevention and treating breast cancer.

## 1. Introduction

The association between metastasis and immunity is considered a hallmark of cancer [1]. Cancer cell metastasis and invasion of vital organs are implicated in poor prognosis and cancer-related deaths [2]. As the first line of defense, the anticancer immune system can distinguish and remove these cancer cells in patients with malignancy. This mechanism initiates T-cell activation, controlled by T-cell receptor (TCR) mediated-signaling pathways, and maintains the immune system homeostasis [3]. However, it has become evident that tumor cells elude immune surveillance for enhancing their progression and metastasis. Tumor utilizes various molecular mechanisms; one of them is the typical immune-suppressive tumor microenvironments that weaken the immune response, allowing an uncontrollable proliferation of cancer cells. More importantly, cancer cells acquire mesenchymal phenotypes that can induce immunosuppression via Epithelial-to-Mesenchymal Transition (EMT).

For epithelial cells, the process of EMT is essential for driving various progressive aspects such as embryonic development and wound healing. However, EMT is also playing a crucial role in immunosuppression and the development of tissue fibrosis, carcinogenesis, metastasis, and drug resistance [4,5]. EMT enhances the migration of epithelial cells to new locations by promoting new organized characters under normal conditions. In contrast, the activated EMT cells in cancer undergo Mesenchymal-to-Epithelial Transition (MET), the crucial phenotype in developing cancer metastasis [6]. In the stage of EMT, the tumor utilizes various molecular mechanisms to escape the immune surveillance; one of them is the upregulation of programmed death-ligand 1 (PD-L1) expression [7]. In various types of cancer, the activation of EMT signaling seems to be a central oncogenic mechanism that upregulates PD-L1 expression [8]. A recent study has cited the close association between EMT and PD-L1, suggesting a bidirectional regulation between EMT status and PD-L1 expression, which leads to tumor immune escape [9,10]. During this process, the cells lose essential epithelial proteins (such as E-cadherin, claudins, cytokeratin, occludins, mucin-1, desmoplakin, and γ-catenin) while express mesenchymal phenotype characteristics with Vimentin, N-cadherin, fibronectin, fibroblast-specific protein 1 (FSP-1), Vitronectin, and smooth-muscle actin which cause immunosuppression and enhance tumor dissemination and migration [11]. Based on that, Pasquier and others have classified the potential therapeutic targets in EMT development into three main groups; the effectors (such as E-cadherin and vimentin), the regulators (such as Zeb, Twist, and Snail transcription factors), and the third group is the inducers that include members of the transforming growth factor-beta (TGF-β) [12].

Immunotherapy has become a novel approach for cancer therapy [13,14]. In advanced cancer, various immune checkpoint inhibitors, including programmed cell death 1 (PD-1), its ligand PD-L1, and cytotoxic T-lymphocyte-associated antigen 4 (CTLA-4), have achieved promising oncological improvements [15,16,17,18]. The ligand PD-L1 (also known as B7-H1/CD274) is a transmembrane glycoprotein encoded by the CD274 gene [19]. PD-L1 has limited expression on a wide variety of normal cells, including B cells, vascular endothelial cells, epithelial cells, macrophages, and myeloid dendritic cells [7,20,21]. However, cancer cells may possess elevated levels of PD-L1, which indicates the significance of inhibiting this ligand. In a previous study, it was noted that BC cells arising from epithelial carcinoma expressed low levels of PD-L1. The opposite was found in its counterpart, which arises from mesenchymal carcinoma cell models that demonstrating high levels of PD-L1 [22].

It was also known from previous studies using BC cell lines that polyphenols have the potential to impair BC metastasis through numerous mechanisms such as activating the tissue inhibitors of metalloproteinases (TIMPs) expression while inhibiting the matrix metalloproteinase (MMPs) expression [23,24,25], interfering with various signaling pathways, including phosphoinositide 3-kinases/protein kinase B/mammalian target of rapamycin (PI3Ks/AKT/mTOR) [26,27], mitogen-activated protein kinase (MAPK) [28,29], Vascular endothelial growth factor (VEGF) [30], nuclear factor kappa light chain enhancer of activated B cells (NF-κB) [31,32,33] pathways, and modulating EMT process. Extensive studies have shown the impact of different polyphenols on EMT signaling pathways [34,35]. However, meager studies have examined polyphenols’ role in inhibiting PD-L1 to modulate breast cancer (BC) cells’ dissemination and metastasis. Therefore, in this review, we emphasized the polyphenol ability to inhibit EMT and PD-L1 activation to identify new options targeting BC metastasis.

## 2. PD-1/PD-L1 Checkpoint in Cancer

Cancer cells have direct mechanisms to suppress anticancer immune signaling. However, another indirect mechanism was also protecting the tumor from immune cell-lined death [3]. This mechanism is orchestrated by the CD28 family of receptors that include the PD-1 receptor, in addition to CD28, cytotoxic T-lymphocyte–associated antigen 4 (CTLA-4), inducible co-stimulator (ICOS), and B- and T-lymphocyte attenuator (BTLA) receptors [36,37,38,39]. Normally, the surface protein PD-1 is expressed on various cells, including monocytes, T cells, B cells, dendritic cells (DCs), and natural killer (NK) cells, and its persistent expression is speculated to maintain the functional silence of T cells through delivering inhibitory signals [40]. The bond between PD-1 and its two ligands PD-L1 and PD-L2 generates either a co-stimulatory immunological synapse or inhibitory signals that inhibit T cell response [41]. Although PD-L2 possesses a higher affinity to PD-1 than its counterpart PD-L1, its relatively low expression leads to non-significant interaction with PD-1, which assigning PD-L1 as the primary contributor of PD-1’ suppressive function [17,19,42]. Hence, PD-1/PD-L1 binding is the crucial mechanism in sustaining the immune-suppressive cancer microenvironment. Although these two proteins can also be expressed under normal physiologic conditions, PD-1 and PD-L1 are considered markers of a compromised immune stimulation as their expression is an indicator of T cell dysfunction [43]. Thus, as the main function, their interaction inhibits cytokine production and T cell activation to retain a consistent immune response [19,44].

In cancer cells, the transcription upregulation of PD-L1 is influenced by various elements. Some of them are summarized in Figure 1. Many cytokines were found to induce PD-L1 expression; however, interferon-gamma (IFN-γ) is the main stimulator along with its IFN-γ and toll-like receptor (TLR) ligands [45,46], which also impair the immunity of effector tumor cells [47]. The cytokine IFN-γ is secreted by various types of cells such as activated lymphocytes [48], T cells [49], B cells [50], macrophages [51], monocytes [52], and dendritic cells [53]. As an immunomodulatory agent, IFN-γ acts as a critical coordinator of the immune response with an anticancer effect [54]. A previous study suggested the close association between the loss of IFN-γ pathway genes—Janus kinases (JAK)1 and JAK2—and the increased resistance to PD-1 blockade immunotherapy [55]. Also, it was shown that the prolonged signaling of IFN-γ coordinates the resistance to immune checkpoint blockade, both PD-L1-dependent and independent [56].

## 3. Oncogenic Signaling Pathways Regulating PD-L1 Expression

### 3.1. MAPK Pathway

The signaling pathway MAPK—also known as extracellular signal-regulated kinases (ERK) and includes rat sarcoma (Ras), rapidly accelerated fibrosarcoma (Raf), mitogen-activated protein kinase kinase (MEK)-MAPK proteins—is a crucial regulator for vital cellular functions such as cell survival, proliferation, and apoptosis [57]. However, aberrant activation of this pathway is detected in about 50% of cancer patients and was associated with cancer initiation and progression [58]. Also, in triple-negative breast cancer (TNBC) patients, MAPK pathway activation endorses immune evasion, leading to chemotherapeutic drug resistance and poor survival rate [59,60]. The MAPK pathway has been demonstrated to control PD-L1 expression in many cancer cells [61]. Remarkably, in TNBC cells, inhibition of this signaling pathway was found to upregulate IFN-γ–induced PD-L1 expression, both in vivo and in vitro studies, whereas inhibiting MAPK and PD-1/PD-L1 was found to synergize the immune checkpoint inhibitors [62]. On the contrary, in BC cells, the interaction of PD-L1/PD-1 stimulates phosphorylation of MAPK, leading to the activation of MAPK pathways and increases the expression of multidrug resistance protein 1 (MDR1) (also known as permeability glycoprotein, P-gp) [63]. Indeed, the MDRI protein is a member of the adenosine triphosphate (ATP)-binding cassette transporter protein superfamily encoded by the ATP binding cassette subfamily B member 1 (ABCB1) gene [64]. In normal tissues, MDRI is usually disseminated to protect the susceptible organs from toxic substances. However, in multidrug-resistant cancer cells, MDRI is upregulated as a challenging mechanism to decrease these drugs’ intracellular concentration. PD-L1 upregulation is closely associated with MDR1 expression in BC cells, and it is mediated by the activation of PI3K/AKT and MAPK signaling pathways [65].

### 3.2. PI3K/PTEN/Akt/mTOR Pathway

Various signaling pathways are involved in IFN-γ-mediated PD-L1 induction [66,67,68,69,70]. However, the process is mainly controlled by the loss of phosphatase and tensin homolog (PTEN) tumor suppressor protein and the consequential oncogenic activation of PI3Ks/AKT/mTOR) pathway [71,72]. The fact that interpreted the decrease in PD-L1 expression after using the AKT inhibitors [73]. In BC, abnormalities in the PI3K/AKT/mTOR pathway are the most frequent genomic defects that affect immune surveillance through the regulation of PD-L1. In TNBC, PTEN loss is associated with estrogen receptor (ER)/progesterone receptor (PR)–negative BC cells [74] and explains the increase of PD-L1 in their MDA-MB-157 cell line model. Also, in the MDA-MB-231 cell line, PTEN knockdown resulted in more significant upregulation in PD-L1 expression than the addition of IFN-γ, the common inducer of PD-L1 expression.

## 4. Transcriptional Control of PD-L1 Expression

### 4.1. The JAK/STAT Pathway

In TNBC, the activation of the signaling pathway JAK/STAT is proportional to the phosphorylated signal transducer and activator of transcription 1/3 (pSTAT1/3), the key transcription factors that significantly regulate cancer cell survival, proliferation, invasiveness, metastasis, and immunosurveillance [75,76,77,78,79,80]. Notably, STATs modify the immune response through various mechanisms, including regulation of PD-L1 expression [18]—when binding to PD-L1 promoter—as indicated by the abolished PD-L1 expression upon their silencing [81]. Moreover, sole inhibition of STAT1 or STAT3 induces a partial downregulation in PD-L1 expression, while a complete downregulation was achieved upon combined inhibition of these transcription factors [82]. Thus, inhibition of JAK/STAT signaling could be a promising therapeutic target in TNBC [83,84].

### 4.2. Hypoxia-Inducible Factor 1α (HIF-1α)

The hypoxic feature is well known in BC and other types of cancer as an adaptive mechanism in the reduced oxygen microenvironment. In response to hypoxia, the activated HIF-1α and HIF-2α [85,86,87] lead to poor prognosis and antiestrogen resistance in BC [88]. Once binding to its hypoxia response elements (HRE) promoter, HIF-1α stimulates the transcription of PD-L1 [89]. Indeed, previous studies revealed the co-existence of upregulated HIF-1α, increased PD-L1 expression, and the attenuation of T-cell function [90,91,92]. In TNBC in-vivo model, the PD-L1 expression level was serving as a biomarker in detecting the level of hypoxia [93]. This finding has advocated inhibitors of HIF-1α/PD-1/PD-L1 as a potential therapeutic target to combat the immune suppression behavior of tumors [94,95,96,97].

### 4.3. NF-ƙB Pathway

The transcriptional factor NF-ƙB has been previously shown to promote and mediate inflammation-cancer pathways, inhibit apoptosis, and enhance tumorigenesis and cancer immune evasion [98,99]. Also, NF-ƙB has the potential to induce PD-L1, either directly through binding to PD-L1 promotor or indirectly, by enhancing the stability of its protein that supports the tumor immune evasion [100]. Notably, the involvement of NF-ƙB in IFN-γ-induced PD-L1 expression has been evidenced by PD-L1 repression in the presence of NF-ƙB inhibitors [98]. Another mechanism that has been previously found to prevent PD-L1 degradation is TNF-α-mediated NF-ƙB activation through enhancing the fifth element of the constitutive photomorphogenesis 9 (COP9) signalosome5 (CSN5) protein [101]. Furthermore, a study on BC demonstrated the ability of natural compounds to induce PD-L1 expression through histone deacetylase 3 (HDAC3)/p300)-mediated NF-ƙB signaling pathway [102]. NF-ƙB-mediated PD-L1 induction is also impacted by aberrant expression of some oncogenes such as B cell lymphoma 3 (Bcl3) [103] and Mucin1 (MUC1) [99] that integrate a variety of signaling pathways. Indeed, Bcl3 promotes IFN-γ-stimulated PD-L1 expression through NF-ƙB p65 acetylation [104]. In TNBC cells, PD-L1 upregulation was revealed to be MUC1-dependent [40]; meanwhile, MUC1 drives PD-L1 overexpression involves MYC proto-oncogene, bHLH transcription factor (Myc), and NF-ƙB-dependent mechanisms [99].

In immune and cancer cells, the Toll-like receptor (TLR)-mediated signaling pathway is a well-known mechanism that upregulates PD-L1 [67] through increasing NF-ƙB activation, which in turn leads to PD-L1 upregulation [105]. Meanwhile, IFNs have been demonstrated to regulate PD-L1 expression on both tumor and non-tumor cells; IFN-γ stands out as the most inducer [45,69]. Also, IFN-γ stimulates nuclear translocation of NF-ƙB signaling pathway, thus upregulating PD-L1’s promoter activity [106].

## 5. PD-L1 Expression in Breast Cancer

It has become evident that overexpression of PD-L1 protects malignant cells from immune detection in various types of cancers, including BC, an event that leads to the increase of tumor aggressiveness and poor disease prognosis [107,108,109,110,111,112,113,114]. In the highly metastatic TNBC subtype, PD-L1 expression is strongly linked to various adverse aspects of aggressiveness, such as advanced cancer grade, lack of ER, and increased infiltration with T-regulatory cells [73,114]. Notably, the significant overexpression of PD-L1 in MDA-MB-231, the typical TNBC cell model [73], leads to tumor escape from the immune system and the worse outcome [9,10]. PD-L1 expression is induced upon EMT activation, and it is closely associated with the mesenchymal features, as clearly manifested in the claudin-low BC, the subtype that is highly associated with poor prognosis and enriched in EMT features [10,115]. More investigations demonstrated the mechanism underlying EMT-mediated PD-L1 upregulation and attributed this aggressive mechanism to the surface markers as shown by PD-L1 concomitant parallel association with CD44 upregulation and CD24 downregulation [10]. In TNBC cells, the PTEN/PI3K pathway is significant in regulating PD-L1 expression. As mentioned earlier, PTEN loss is a mechanism that promotes PD-L1 expression through the PI3K/AKT/mTOR pathway, and it is correlated with ER/PR–negative tumor [74,116]. Moreover, glycosylation inhibitors were significantly linked to the repressed PD-L1 expression in BC cells [117] and momentous purge of TNBC cells [118]. For example, in TNBC cells, the upregulation of PD-L1 expression and the activation of NF-ƙB is transmembrane glycoprotein MUC1-dependent [99]. MUC1 is overexpressed in various types of cancer and implicated in multiple signaling pathways that enhancing cancer growth and maintenance [119]. The enforced TGF-β1 upregulation was found to induce PD-L1 expression in normal breast cells. However, the mechanism was mainly driven by the induced EMT, not TGF-β1 itself [10]. Overall, these findings support the rationale for applying therapeutic approaches targeting the PD-1/PD-L1 via PI3K pathway in TNBC metastatic subtype [73].

## 6. Epithelial-to-Mesenchymal Transition (EMT) Markers Mediating PD-L1 Induction in Breast Cancer

It is well known that the pro-metastatic phase within the tumor microenvironment is linked to inflammation. Indeed, the host’s tumor-infiltrating immune cells secrete various types of cytokines and chemokines such as TGF-β in an endeavor to fight cancer [120]. Unfortunately, this mechanism provokes the EMT process and promotes cancer cell invasion and migration [121,122]. Contrary to the common belief, many studies using in vivo and in vitro BC models have evidenced upregulated expression of PD-L1 along with normal PTEN and the lack of the INF-γ [10]. Hence, the existence of another mechanism underlying the regulation of PD-L1 in BC was suggested [10]. Indeed, in some types of cancer, a poor prognosis was found in PD-L1(+)/EMT (+) compared with the PDL1(+)/EMT (−) one, which indicates the importance of targeting EMT to limit cancer migration and prognosis [123]. A recent study has summarized the involvement of different molecules such as MUC1, TGF-β, and NF-ƙB [10,99] in EMT-mediated PD-L1 upregulation in BC [115]. Other oncological studies cited the opposite, and they revealed the importance of PD-L1 signaling in maintaining EMT status [10,124,125,126] (Figure 2). Nevertheless, both mechanisms will eventually lead to tumor immune escape [10,115]. Thus, EMT status was considered a co-biomarker with PD-L1 to speculate the prognosis and the likelihood of response to PD-1/PD-L1 checkpoint blockade [115].

Meanwhile, PD-L1 can be stimulated directly; another profound indirect mechanism underlying EMT-mediated PD-L1 expression was demonstrated [10]. A significantly upregulated level of PD-L1 was attributed to the tumor cell surface markers as mentioned above [10]. Indeed, EMT -mediated PD-L1 expression is highly suggested in the claudin-low subtype of TNBC, characterized with high EMT features [10], while decisively downregulated PD-L1 reversed EMT process, which strongly suggests the important role for PD-L1 targeted therapy in this subset of the disease [10]. Also, TGF-β cytokine was the primary inducer of EMT [120], which augments the expression of PD-L1 in BC cells [10]. Interestingly, the upregulation of PD-L1 in BC was attributed to EMT activation but not TGF-β itself [10].

The role of EMT transcription factor (EMT-TFs) in controlling PD-L1 expression was also revealed. It was previously suggested that EMT-TFs, including Zeb, Twist, and SNAIL family proteins, mediate EMT regulation and tumor progression [4,127] and bridge the link between inflammation and cancer [127,128]. Moreover, higher expression of Zeb1, Snail, N-cadherin, and Vimentin and low expression of E-cadherin were closely correlated with the upregulated level of PD-L1 [115,129]. In TNBC cells, various transduction signaling pathways were involved in EMT-mediated PD-L1 expression, with the MAPK pathway the most crucial one [130,131]. Various examples also were reported for proteins involved in the EMT process. The overexpression of the insulin-like growth factor 1 receptor (IGF1R) and focal adhesion kinase (FAK) signaling was crucial for EMT and metastasis [130]. These signaling pathways caused an enhancement of the mesenchymal markers’ expression, Zeb1, Snail1, and Vimentin, while a reduction of the epithelial markers claudin-1, E-cadherin, and Zonula occludens-1 (ZO-1) was found. Similarly, adapter molecule Crk (Crk) protein is implicated in various signaling pathways regulating EMT and EMT-stimulate PD-L1. The Crk mechanism for enhancing cancer metastasis was achieved by upregulating the expression of Zeb1 and N-cadherin and repressing E-cadherin levels [129,132,133]. Indeed, targeting signaling pathways and cytokine-induced EMT could hold promises in inhibiting BC cell dissemination and metastasis [134,135,136,137,138,139,140].

On the other hand, a growing body of literature has suggested the implication of upregulated EMT in increasing drug resistance and cancer progression. This resistance behavior was exhibited in patients diagnosed with solid cancers, including BC, presenting a considerable challenge [15,141]. Indeed, EMT was associated with the upregulated expression of many (ATP)—binding cassette (ABC) transporters that ultimately lead to multidrug resistance [120,142,143]. Hence, combining therapeutic agents against EMT-TFs was a promising approach to overcoming these tumors’ resistance mechanisms [144].

## 7. Breast Cancer Treatment

For decades, cytotoxic chemotherapeutic drugs were the standard medical treatment for BC patients [145]. Various target—directed approaches have evolved to treat and manage the heterogenous BC characterized by diverse molecular subtypes and stages [145]. Chemotherapeutics drugs with cytotoxic effects—doxorubicin and paclitaxel—are typically applied for patients with metastasized BC. Other treatments, including gemcitabine, cisplatin derivatives, 5-fluorouracil, or vinorelbine, are also used. On the other hand, combined treatments with chemotherapy drugs are considered a promising approach for enhancing BC therapy outcomes [146]. For instance, the estrogen antagonists—tamoxifen or fulvestrant—combined with the aromatase inhibitors—anastrozole, letrozole, and exemestane—are used in the hormone-dependent (ER+/PR+) BC cells [147]. Also, bevacizumab, a monoclonal antibody therapeutic, targets vascular endothelial growth factor receptor (VEGFR), hindering the angiogenesis pathway [148,149,150]. Another monoclonal antibody, trastuzumab, could be used in patients overexpressing the HER-2 receptor, combined with therapeutic hormonal drugs such as the selective HER-2 pathway inhibitors lapatinib [147,149,150]. Moreover, various emerging drugs have shown the potential to overcome hormonal therapy resistance when combined with hormonal drugs [145]. These included the cyclin-dependent kinases 4 and 6 (CDK4/6) inhibitors such as abemaciclib, palbociclib, and ribociclib [151]—which impact cell cycle progression—and inhibitors of PI3K/AKT/mTOR pathway such as buparlisib, pictilisib, pilaralisib, and voxtalisib [152,153]. On the contrary, TNBC—the most aggressive cells with abolished biomarkers expression—the classical chemotherapeutic drugs, such as taxanes, anthracyclines, and platinum agents, remained the exclusive therapeutic option [148,149,150], and they are currently used with/without the monoclonal antibody against VEGF bevacizumab [154]. Recently, new targeted drugs were introduced and still undergo clinical trials for optimizing BC therapeutic outcome, including poly adenosine diphosphate (ADP)-ribose polymerase (PARP) inhibitors—olaparib, talazoparib, veliparib, niraparib, and rucaparib—for those with mutated breast cancer type 1/2 susceptibility protein (BRCA1/2) [155,156,157], the antibody-drug conjugate Glembatumumab vedotin, the androgen receptor inhibitor bicalutamide, and the anti-PD-1 monoclonal antibody pembrolizumab.

## 8. Current Breast Cancer Immunotherapeutic Strategies

While the treatment regimens of BC have greatly improved in recent years, the disease’s emerging subtypes raised a significant challenge that classified BC as the most frequent cancer type affecting women [158]. TNBC cells—a BC subtype lacking the expression of ER, PR, and the overexpression of the humane epidermal receptor (HER)—were further classified into basal-like and claudin-low subtypes [159,160,161]. The lack of hormonal receptors in TNBC urged the need for developing new therapeutic approaches targeting these subtypes [10,162]. Hence, cancer immunotherapy is considered a narrative approach in different types of cancer [13,163]. Various immune checkpoint blockade, mainly PD-1 and its ligand, PD-L1—the most prognostic biomarker—and CTLA-4 inhibitors, have been established in the clinics [164,165]. Fortunately, PD-1 and PD-L1 inhibitors have been promising in treating various kinds of cancer, including BC [166]. From 2011-2017 exhibited the emergence of valuable drugs that inhibit PD-1 (Pembrolizumab and Nivolumab) and PD-L1 (Atezolizumab, Avelumab, and Durvalumab), as well as the monoclonal antibody Ipilimumab that targeting CTLA4 [167,168].

Although the PD-1 and PD-L1 blockade immunotherapy has achieved an incredible clinical outcome in some subsets of BC patients [169], so far, PD-1 blockade works only in PD-1(+)/PD-L(+) but not in PD-1(−) patients [8,18,170]. Meanwhile, not all PD-L1-expressing cancer patients responded to PD-1/PDL1 inhibitors; PD-L1(−) tumors may respond to these agents [171]. Most importantly, using the immunotherapeutic candidates —targeting PD-L1/PD-1 pathway—was found to enhance other antitumor treatment approaches [172]. For instance, in BC tumor, a solely administered doxorubicin, the conventional chemotherapy drug, attenuated PD-L1’s cell surface expression and exhibited apoptotic effect; however, it increased PD-L1 nuclear expression [172]. Furthermore, the co-existence of doxorubicin and PI3K/AKT pathway inhibitor abolished the doxorubicin-induced nuclear up-regulation of PD-L1, suggesting the significant role of the PI3K/AKT pathway in the nuclear upregulation of PD-L1 in BC cells [172,173].

## 9. Polyphenols and Cancer

Recently, special attention has been directed to the polyphenols found in a wide variety of edible plants, including vegetables, fruits, soy products, in addition to cereal, wine, and tea [174,175]. Myriads of epidemiological studies have cited the uses of polyphenol in treating a diversity of health issues, including infection [176,177], inflammation [178], oxidative stress [179], bone diseases [180], cardiovascular disease [181], and cancer [182]. In cancer research, extensive literature has correlated the consumption of polyphenol-rich food and the lower risk of cancers [183,184,185,186,187,188,189,190]. It has been suggested that polyphenols may inhibit tumors at various stages, including initiation, relapse, progression, and metastasis to other organs [191,192,193]. The well-known antioxidant activities of the polyphenols were found to induce a chemopreventive effect [192], together with their anticancer effect that conveys antioxidant-independent mechanisms [192,194].

Polyphenols have demonstrated a vital role in modulating various signaling pathways and modifying proteins-mediating cancer progression [34,35,195]. Indeed, polyphenols exhibited anti-oncogenic effects in the NF-ƙB transcription factors, Wnt/β-catenin, peroxisome proliferator activator receptor-gamma (PPAR-γ), STAT3, nuclear factor erythroid 2 (Nrf2), sonic hedgehog (Shh), activator protein-1 (AP-1), growth factors receptors (epidermal growth factor receptor, EGFR; Erb-B2 receptor tyrosine kinase 2, ErbB2, VEGFR; insulin like growth factor1 receptor, IGF1-R). Polyphenols also have been shown prospectively to reverse EMT-underlying tumor metastasis by modifying miRNA’s expression [6]. Moreover, these compounds revealed anti-inflammatory effects through modulating the pro-inflammatory mediators, tumor necrosis factor-α (TNF-α), interleukins (ILs), Cyclooxygenase (COX)-2, 5-Lipoxygenases (LOX), and various protein Kinases (PI3K, mTOR, AMPK, Bcr-abl, and Ras/Raf) [34,35,195,196].

Although many dietary polyphenolic compounds have shown various pharmacological effects, there are still challenges that should be considered for many other polyphenols to be effective in clinical practices [197]. When taking orally—since the mouth is the most common route of administration for small molecule drugs and nutraceuticals [198]—these polyphenols might face many obstacles before reaching their site of action. The challenges may include poor aqueous solubility, weak oral absorption, low bioavailability, or fast systemic elimination [197]. To manage the pharmacokinetics profile of such perplexing polyphenols, various formulations could be approached. Many developed formulations have already been pharmaceutically applied to manage these barriers, such as nanogels, nanoparticles, nanospheres, liposomes, complexation, micelles, and solid dispersions [199]. Significantly, interactions with other elements found in food and other drugs might be highly anticipated with some polyphenols [6], even though they could be prevented by specialized formulations, avoiding specific food intake, and managing dosage regimens.

Clinical trials in BC patients evidenced the potential of the dietary polyphenolic compounds to increase apoptosis while decreasing various tumor biomarkers [200,201], including steroid hormones [202,203], carcinoembryonic antigen (CEA), VEGF [204], and radiation dermatitis severity score (RDS) [205], in addition to anti-inflammatory effects [206]. On the other side, none of these studies demonstrated the potential of these polyphenols to modulate the immune response in BC patients. 

Nevertheless, there is a continuous interest in investigating dietary flavonoids as antitumor immunity agents [207,208]. Here, we summarized the most-studied compounds and highlighted their potential to target PD-L1 in BC cells, either directly or indirectly, through modulating EMT markers-mediating PD-L1 activation. This summary will also provide a closer look at the polyphenols’ most specific studies that could be used combined with the current use of PD-L1 blockade and anti-PD-1 immunotherapy to enhance their efficacy against BC.

### 9.1. Curcumin

Curcumin is a natural polyphenol compound extracted from the turmeric roots and used for a long time as a traditional medicine in the Ayurveda [209,210,211]. This compound has demonstrated various pharmacological properties, including antioxidant [212], anti-inflammatory [213], antimicrobial [214], immunomodulatory [215], and hepatoprotective [216] properties. Furthermore, curcumin has shown anti-metastatic effects through targeting various intracellular signaling pathways implicated in PD-L1 upregulation [217], including NF-ƙB [218], MAPK [219], Wnt/β-catenin [220], PI3K/Akt/mTOR [221], hedgehog [222], Notch [223], and block IκB kinase (IKK) activity that consequently inhibits NF-ƙB signaling pathway [224,225] (Figure 3).

In BC, curcumin also was a potent agent in targeting various genes mediating the EMT process. For instance, it targets H19 long non-coding RNA. Upregulated H19 promotes EMT through increasing vimentin expression and repressing E-cadherin expression, and more importantly, contributes to tamoxifen-resistant tumors [226]. The highly expressed H19 also plays a crucial role in various cellular events, including proliferation, chemoresistance, endocrine resistance, migration, invasiveness, and metastasis [227,228]. Furthermore, recent cohort studies have evidenced the close association between the long non-coding RNA and PD-L1 expression [229,230]. Thus, targeting this gene was considered a key in PD-L1 inhibition [231].

Upon curcumin exposure, other target proteins associated with EMT and metastasis—slug, β-catenin, receptor tyrosine kinase (RTK, aka; AXL), CD24, and vimentin—were repressed in the MDA-MB-231 cells model of TNBC [222,232,233]. These proteins are upregulated in both human and murine BC [234,235,236]. Moreover, curcumin was also found to impact TGF-β and PI3K/AKT signaling pathways regulating doxorubicin-stimulated EMT activation [237,238]. The intrinsic β-catenin is a crucial oncogenic protein in driving cancer initiation and progression through modulating the transcription of many genes such as slug -mediating BC metastasis. Thus, the inhibition of β-catenin, hindering the trans-stimulation of slug and, consequently, restores E-cadherin expression of epithelial phenotype [239,240]. The oncogene β-catenin is a well-known regulator of PD-L1–mediated immunosuppression as revealed by the significant abolition in PD-L1 expression upon reducing β-catenin. On the contrary, upregulated β-catenin was positively correlated with the increased level of PDL1′s protein expression [8,241]. Also, upon inhibiting Axl kinase, a significant decrease in tumor growth was found in the mouse models, the effect that was further potentiated when combined with PD-1 blockade [236]. Pharmacological repression of Axl activity was found to decrease the mRNA expressions of PD-L1, the finding that revealed the implication of Axl in regulating PD-L1 protein expression [242]. Moreover, the cytokine TGF-β—as another mediator in EMT development—is involved in many cellular events and upregulating the expression of PD-L1 [243]. TGF-β has a tumor promoter role in the advanced stages of the disease as it enhances EMT and metastasis [244,245,246]. An interesting report on combining TGF-β inhibitors with PD-1/PD-L1 immune checkpoint blockade has revealed a tumor regression [247]. Therefore, the pharmacological modulation of β-catenin, Axl, and TGF-β are considered putative trends in cancer therapy [248]. This information strongly suggests the pivotal role of curcumin in inhibiting PD-L1 directly and indirectly through deactivating EMT markers in BC patients and ultimately limiting metastasis.

### 9.2. Apigenin

The flavone apigenin is found in various fruits, vegetables, and herbs, such as parsley, onions, grapefruit, oranges, and chamomile [249,250]. Apigenin was previously found to demonstrate various biological activities, including antioxidant [251], anti-inflammatory [252], antibacterial, antiviral [253], and anticancer effects [254]. Fortunately, apigenin is considered a safe compound for normal healthy cells [255]. As an anticancer agent, low concentrations of apigenin were found to inhibit the proliferation, while a significant apoptotic effect was induced at higher concentrations of the compound [255,256,257,258,259]. Moreover, apigenin’s anti-metastatic effect has been revealed in several cancers, including BC [260,261,262]. The compound showed immunomodulatory properties by targeting the PD-1/PD-L1 checkpoint as a promising immunotherapy candidate [17]. The ability of apigenin to inhibit PD-L1 was also investigated in human and mouse mammary carcinoma cells. 

Apigenin was found to inhibit IFN-γ-induced PD-L1 upregulation in MDA-MB-468 TNBC cells, HER2+SK-BR-3, human mammary epithelial cells, and 4T1mouse mammary carcinoma cells. In MDA-MB-468 and 4T1 cells, the repression of PD-L1 level was associated with reduced phosphorylation of STAT1 [263] (Figure 4). Luteolin, the major metabolite of apigenin, was also found to inhibit IFN-γ-induced PD-L1 expression in MDA-MB-468 cells. In the MDA-MB-231 TNBC cell, apigenin did not repress PD-L1 expression, and its anti-metastatic effect was not directed to the EMT markers, Vimentin, or N-cadherin. Instead, the compound repressed IL-6-mediated EMT signal-linked N-cadherin expression [264]. Certainly, the positive association between IL-6 and EMT in the tumor microenvironment has been proven in various types of cancer, including BC, leading to cell migration and invasiveness [265,266,267,268]. It is collectively suggested that more investigations are needed to characterize the effects of apigenin on EMT and PD-L1 inhibition as a safe immunotherapeutic candidate drug for specific subsets of BC disease.

### 9.3. Hesperidin

Hesperidin is a flavonoid found in various Rutaceae family members [269] and was used in China as traditional herbal medicine. In pharmacological studies, the compound exhibited various anticancer effects with anti-proliferative [270], anti-inflammatory [271], and apoptotic [269,272] properties. Previously reported research indicated the safety of the compound against normal cells. Remarkably, the properties of hesperidin endorsed the compound’s use as an anticancer candidate against BC and other cancer types. The significant effect of hesperidin in inhibiting the expression of EMT markers was recently exposed [273]. Also, one report investigating the effect of hesperidin in MDA-MB-231 cells has cited its ability to inhibit the levels of PD-L1 at both the protein and the transcriptional level through inhibiting PI3K/Akt and NF-ƙB signaling pathway [274] (Figure 5). These previous findings support our hypothesis that this polyphenol compound has the potential to ultimately inhibit PD-L1, directly or indirectly, through impacting EMT markers. Still, more emphasis on hesperidin and its mechanism against EMT markers and PD-1/PD-L1 checkpoints are highly encouraged.

### 9.4. Resveratrol

Resveratrol is a polyphenol component found in peanuts and grapes as well as other plants [275]. The compound has shown significant biological activities and may hold promises as a therapeutic agent against cancer [276]. Notably, resveratrol showed a potency to inhibit various tumors’ initiation and progress [276,277]. The impact of this polyphenol in BC is multidisciplinary as extensive studies revealed resveratrol’s ability to utilize different mechanisms in targeting epigenetic response, cell proliferation, apoptosis, EMT/metastasis, and most appreciably, increased sensitivity to chemotherapy [278]. In BC, resveratrol mediates cellular aging and inhibits the EMT process by inducing the tumor suppressor Rad9-dependent mechanism [279]. The adaptor protein Rad9 is crucial for the DNA damage response (DDR) protein [280]. A reduction of Rad9 expression was detected in the highly invasive MDA-MB-231 cells [279]. Most importantly, Rad9 protein has shown a selective mechanism in controlling genes linked to EMT, such as p21 [281], Neil1 [282], and slug [280]. Also, the compound inhibited cell migration through PI3K/Akt and Wnt/β-catenin signaling pathways [283] (Figure 6)—the pivotal elements in regulating PD-L1 protein expression—in BC cells [284]. Remarkably, the compound demonstrated a potential to overcome chemotherapy resistance in BC. Resveratrol sensitized the cells to tamoxifen through TGF-β/Smad-driven EMT [285] and promotes cell sensitization to doxorubicin by inhibiting EMT and modulating SIRT1/β-catenin signaling pathway [285,286]. Similarly, recent studies using MDA-MB-231 cells indicated resveratrol’s ability to inhibit cell migration by reversing TGF-β1-induced EMT [140] and inducing a significant suppression of PD-L1 through targeting PD-L1 glycosylation enzymes [287]. Therefore, resveratrol’s unique properties should be highlighted in the field of BC immunotherapy and drug resistance management.

### 9.5. Sativan

The compound (−)-sativan is a natural isoflavone found in *Spatholobus suberectus.* The traditional Chinese herb *Spatholobus suberectus* is commonly used in China for treating many diseases, including anemia, rheumatism, and menoxenia [288]. This herb has been found to possess antioxidant and anti-inflammatory properties [289]. Several recent studies indicated the anticancer effects of *Spatholobus suberectus* in BC with the potential to trigger apoptosis, cell cycle arrest, lactate dehydrogenase inhibition [290], and preventing cancer cell migration through the MAPK PI3K/AKT pathway [291] (Figure 7). Further, a recent study has demonstrated the potential of the compound sativan to induce an anticancer effect in TNBC cells through inhibiting both of EMT process and PD-L1 mRNA expression [292]. Sativan impacted various oncogenic transcription regulators mediated EMT activation [292] and showed the ability to stimulate E-cadherin while decreasing N-cadherin and vimentin. Moreover, Snail and slug were significantly inhibited by the compound [292]. As with any other novel therapeutic agent, further investigations should be considered to shed light on the possible therapeutic mechanisms that can be disclosed for the BC immunotherapy approach.

## 10. Conclusions

Cancer metastasis to vital organs is the leading cause of poor prognosis and cancer-related death, and it is accomplished by the immune evasion strategy. Indeed, tumor suppresses the anticancer immune signaling, either directly or indirectly. One mechanism in inhibiting these signals is the upregulation of the CD28 family of the receptor, in particular PD-1. Indeed, the association between PD-1 and its ligand PD-L1 provokes immune inhibitory signals. In BC cells, overexpression of PD-L1 protects malignant cells from immune devastation, and it is strongly linked to tumor aggressiveness, poor prognosis, and drug resistance. In tumors, the EMT defend process drives various aspects of carcinogenesis, metastasis, immunosuppression, and drug resistance. Notably, there is a strong association between activated EMT and PD-L1 expression.

On the other hand, polyphenols have shown a significant effect as elements of anticancer immunity. Indeed, polyphenols can inhibit the PD-L1 expression directly. However, in this review, we highlighted the indirect mechanism of polyphenol in inhibiting EMT-mediate PD-L1 expression through inhibiting the mesenchymal protein and upregulating the epithelial counterpart. Indeed, apigenin and its major metabolite, luteolin, were able to inhibit IFN-γ-induced PD-L1 expression, in addition to repressing IL-6 mediated EMT process. Also, hesperidin was found to impact EMT markers and targeting various signaling pathways such as PI3K/Akt and NF-ƙB signaling pathway. Resveratrol also showed a potential to inhibit the EMT process by stimulating the tumor suppressor Rad9-dependent mechanism, reversing TGF-β1-induced EMT, as well as targeting PD-L1 glycosylation enzymes. Furthermore, sativan has been shown to impact various oncogenic transcription regulators mediated EMT activation through stimulating E-cadherin while inhibiting N-cadherin, Vimentin, Snail, and Slug. In conclusion, having a direct/indirect/or both mechanisms in targeting PD-L1 expression holds promise in limiting metastasis and treating patients suffering from BC disease. Various polyphenolic compounds have been used in BC clinical trials. These compounds have demonstrated promising anticancer effects in patients with various stages of BC. These effects include anti-inflammatory, pro-apoptosis induction, and suppression of various tumor biomarkers such as CEA, VEGF, and RDS. On the other side, few limited studies have proved the potential of these compounds to impact EMT-underlying tumor metastasis through modifying miRNA’s expression. Hence, further comprehensive investigations are suggested to highlight and focus on the potential of these dietary polyphenolics to reverse or inhibit the challenged EMT process. 

## Figures and Tables

**Figure 1 nutrients-13-01718-f001:**
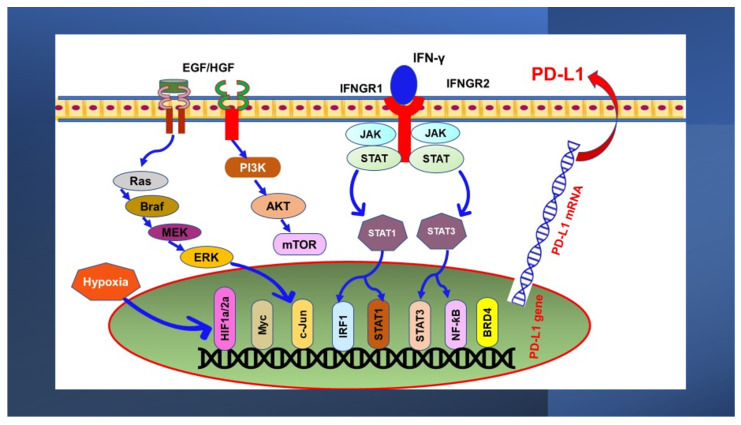
Tumor-intrinsic PD-L1 signaling in cancer initiation and development. The diagram highlights downstream signaling of PD-L1 activation in cancer. Hypoxia-inducible factors, HIF; interferon regulatory factor1, IRF1; MYC proto-oncogene, bHLH transcription factor, Myc; Janus kinase, JAK; signal transducer and activator of transcription (STAT)1/3; nuclear factor-kappa B, NF-ƙB; bromodomain-containing protein 4, BRD4; interferon-gamma, IFN-γ; IFN-γ receptor 1/2, IFNGR1/2; phosphoinositide 3-kinase, PI3K; protein kinase B, AKT; mammalian target of rapamycin, mTOR; extracellular-signal-regulated kinase, ERK; mitogen-activated protein kinase, MEK; B-Raf Serine/Threonine-Protein, BRAF; rat sarcoma, Ras; epidermal growth factor, EGF; hepatocyte growth factor HGF; programmed death-ligand 1, PD-L1.

**Figure 2 nutrients-13-01718-f002:**
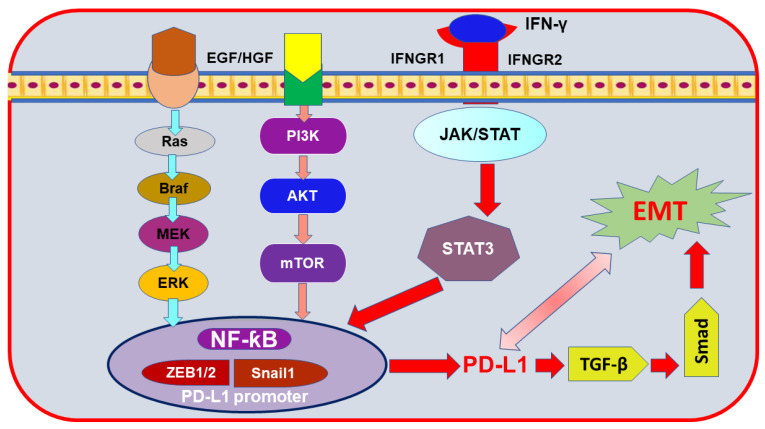
PD-L1-mediated EMT stimulation. The diagram highlights the downstream signaling of EMT in cancer. Interferon-gamma, IFN-γ; IFN-γ receptor 1/2, IFNGR1/2; epidermal growth factor, EGF; hepatocyte growth factor HGF; Janus kinase, JAK; signal transducer and activator of tran-scription3, STAT3; nuclear factor-kappa B, NF-ƙB; phosphoinositide 3-kinase, PI3K; protein kinase B, AKT; mammalian target of rapamycin, mTOR; zinc finger E-box binding homeobox 1/2, Zeb1/2; Snail family transcriptional repressor 1, Snail 1; extracellular-signal-regulated kinase, ERK; mitogen-activated protein kinase, MEK; B-Raf Serine/Threonine-Protein, BRAF; rat sarcoma, Ras; programmed death-ligand 1, PD-L1; transforming growth factor-beta, TGF-β; mothers against decapentaplegic, Smad; epithelial-mesenchymal transition, EMT.

**Figure 3 nutrients-13-01718-f003:**
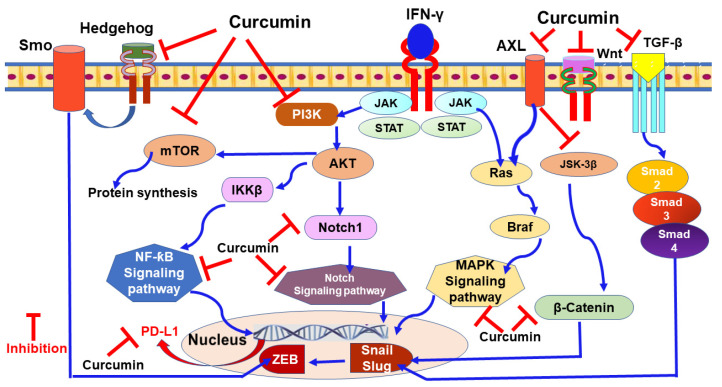
The mechanism of curcumin-mediated programmed death-ligand 1 (PD-L1) inhibition in breast cancer cells. Signal transducer and activator of transcription, STAT; Janus kinase, JAK; nuclear factor-kappa β, NF-ƙB; smoothened, frizzled class receptor, Smo; interferon-gamma, IFN-γ; Phosphoinositide 3-kinase, PI3K; protein kinase β, AKT; mammalian target of rapamycin, mTOR; mitogen-activated protein kinase, MAPK; B-raf serine/threonine-protein, Braf; rat sarcoma, Ras; transforming growth factor-β, TGF-β; mothers against decapentaplegic, Smad; wingless-related integration site, Wnt; zinc finger E-box binding homeobox, ZEB; snail family transcriptional repressor, Snail; inhibitor of kappa light polypeptide gene enhancer in β-cells, kinase beta, IKKβ; programmed death-ligand 1, PD-L1; AXL receptor tyrosine kinase, AXL.

**Figure 4 nutrients-13-01718-f004:**
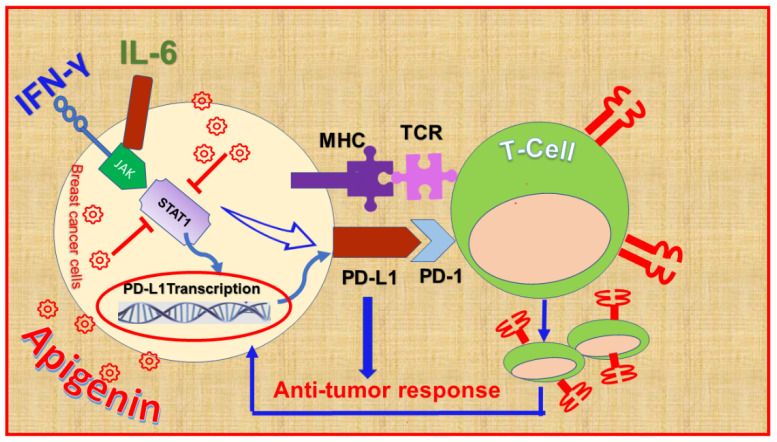
The mechanisms of Apigenin-mediated programmed death-ligand 1 (PD-L1) inhibition in breast cancer cells. Interferon-gamma, IFN-γ; Interleukin 6, IL-6; Janus kinase, JAK; signal transducer and activator of transcription1, STAT1; major histocompatibility complex, MHC; T-cell receptor, TCR; programmed cell death protein 1, PD-1.

**Figure 5 nutrients-13-01718-f005:**
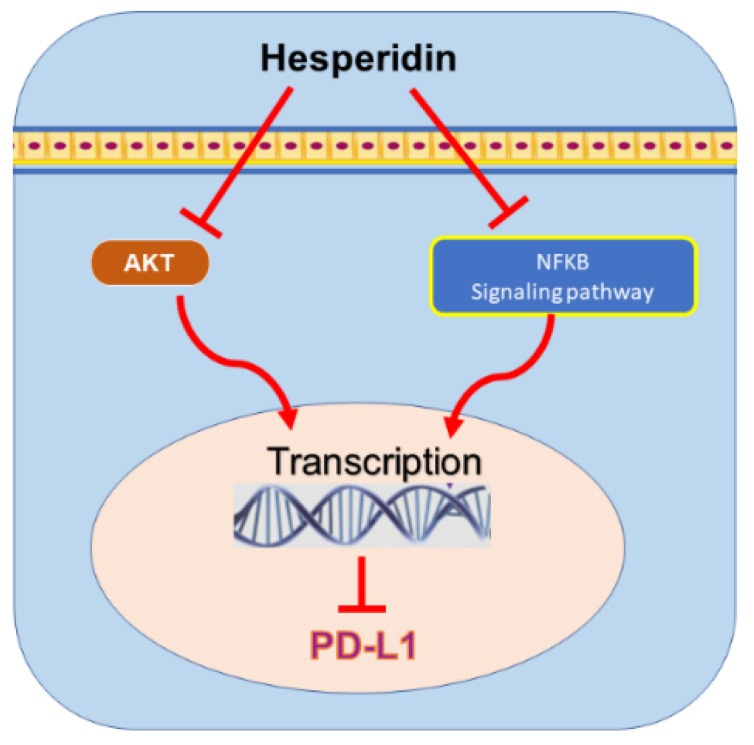
The mechanisms of hesperidin-mediated programmed death-ligand 1 (PD-L1) inhibition in breast cancer cells. Protein kinase B, AKT; nuclear factor-kappa B, NF-ƙB.

**Figure 6 nutrients-13-01718-f006:**
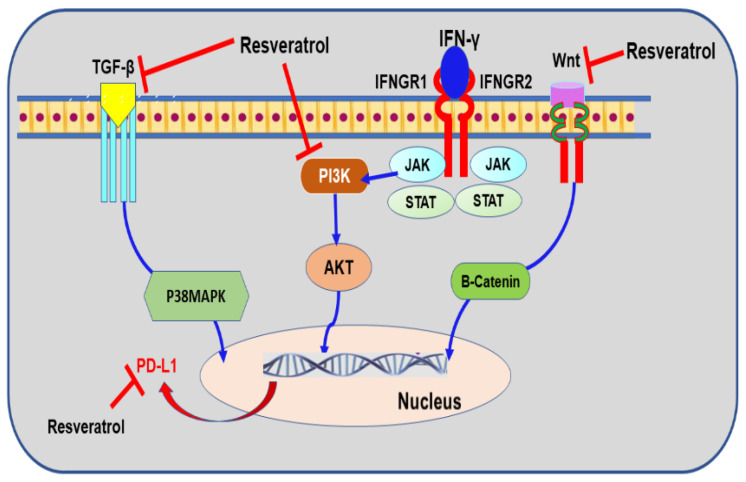
The mechanisms of resveratrol-mediated programmed death-ligand 1 (PD-L1) inhibition in breast cancer cells. Interferon-gamma, IFN-γ; IFN-γ receptor 1/2, IFNGR1/2; transforming growth factor-beta, TGF-β; Janus kinase, JAK; signal transducer and activator of transcription, STAT; phosphoinositide 3-kinase, PI3K; protein kinase B, AKT; mitogen-activated protein kinase, MAPK.

**Figure 7 nutrients-13-01718-f007:**
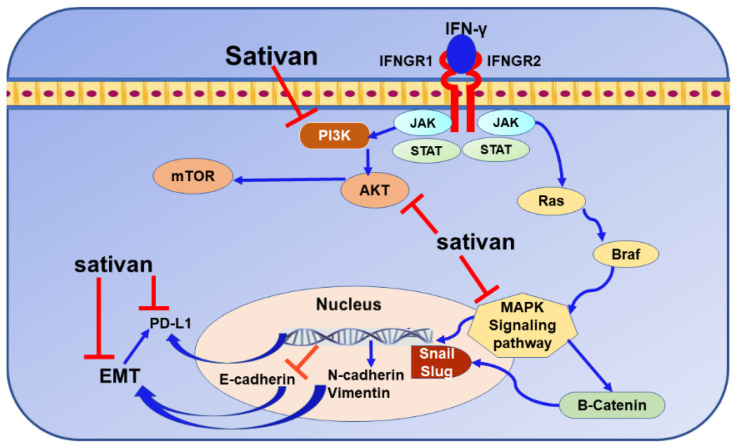
The mechanisms of sativan-mediated programmed death-ligand 1 (PD-L1) inhibition in breast cancer cells. Signal transducer and activator of transcription, STAT; Janus kinase, JAK; Phosphoinositide 3-kinase, PI3K; protein kinase B, AKT; mammalian target of rapamycin, mTOR; mitogen-activated protein kinase, MAPK; B-Raf Serine/Threonine-Protein, BRAF; rat sarcoma, Ras; interferon-gamma, IFN-γ; IFN-γ receptor 1/2, IFNGR1/2; Snail family transcriptional repressor, Snail; Epithelial-to-Mesenchymal Transition, EMT.

## Data Availability

Not applicable.

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
