# Peer review of "Polyphenols Modulating Effects of PD-L1/PD-1 Checkpoint and EMT-Mediated PD-L1 Overexpression in Breast Cancer"

_nutrients, 2021, doi:10.3390/nu13051718_

Round 1

Reviewer 1 Report

Revision manuscript Nutrient-1199660

The authors of the review entitled  “ Polyphenols modulating effects of PD-L1/PD-1 checkpoint and EMT-mediated PD-L1 overexpression in breast cancer cells”  (correct cells instead of cell in the title) deal with an interesting and well-debated topic regarding molecular mechanisms. However, to improve the review the authors should also discuss the limitations in the pharmacological use of polyphenols such as their poor bioavailability.

They should also mention, if there are, human clinical trials of polyphenols they have chosen to treat in this review, as modulators of immune response and EMC transition in breast cancer.

 Moderate English changes required.

Author Response

Response to the first reviewer

Revision manuscript Nutrient-1199660

  1. The authors of the review entitled “Polyphenols modulating effects of PD-L1/PD-1 checkpoint and EMT-mediated PD-L1 overexpression in breast cancer cells” (correct cells instead of cell in the title) deal with an interesting and well-debated topic regarding molecular mechanisms.

Response

Thank you for the suggestion, and the authors revised it in Line 3 and highlighted it in yellow.

  1. However, to improve the review, the authors should also discuss the limitations in the pharmacological use of polyphenols, such as their poor bioavailability.

Response

Thank you for the suggestion. The authors added a paragraph (Line 396-409) and highlighted it in yellow as follows:

Although many dietary polyphenolic compounds have shown various pharmacological effects, there are still challenges that should be considered for many other polyphenols to be effective in clinical practices [198]. When taking orally—since the mouth is the most common route of administration for small molecule drugs [199]—these polyphenols might face many obstacles before reaching their site of action. The challenges may include poor aqueous solubility, weak oral absorption, low bioavailability, rapid metabolism, or fast systemic elimination [198]. To manage the pharmacokinetics profile of such perplexing polyphenols, various formulations could be approached. Many advanced formulations have already been pharmaceutically applied to manage these barriers, such as nanogels, nanoparticles, nanospheres, liposomes, complexation, micelles, and solid dispersions [200]. Significantly, interactions with other elements found in food and other drugs might be highly anticipated with some polyphenols [6], even though they could be prevented by specialized formulations, avoiding specific food intake, and managing dosage regimens.

  1. They should also mention, if there are, human clinical trials of polyphenols they have chosen to treat in this review as modulators of immune response and EMC transition in breast cancer.

Response:

The authors agree with the review and add the following yellow-highlighted sentence (Line 409-415.)

Clinical trials in BC patients evidenced the potential of the dietary polyphenolic compounds to increase apoptosis while decreasing various tumor biomarkers [201,202], steroid hormones production [203,204], the expression of the carcinoembryonic antigen (CEA) and VEGF [205], and radiation dermatitis severity score (RDS) [206], in addition to its anti-inflammatory effects [207]. On the other side, none of these studies demonstrated the potential of these polyphenols to modulate the immune response in BC patients.

  1. Moderate English changes required.

Response:

Thank you for the suggestion. The authors addressed this issue with the use of English.

Language software review.

Reviewer 2 Report

Messeha et al., in their manuscript “Polyphenols modulating effects of PD-L1/PD-1 checkpoint and EMT-mediated PD-L1 overexpression in breast cancer cell” reviewed various phytochemicals used against breast cancer by modulating EMT. Manuscript needs to be improved based on the following comments.

  1. Provide more information present and past approved drugs used for treating breast cancer and their molecular mechanism.
  2. Illustrate a figure showing how PD-L1 expression regulates EMT in breast cancer for better understanding.
  3. Provide limitations and future prospective in conclusion.
  4. Avoid grammatical mistakes and typo errors.

Author Response

Response to the second reviewer

Manuscript needs to be improved based on the following comments.

  1. Provide more information on present and past approved drugs used for treating breast cancer and their molecular mechanism.

Response

Thank you for the suggestion. The authors added the following paragraph (Line 318-345) as follows.

  1. Breast cancer treatment

For decades, cytotoxic chemotherapeutic drugs were the standard medical treatment for BC patients. [146]. Various target-directed approaches have evolved to treat and manage the heterogenous BC characterized by diverse molecular subtypes and stages [146]. Chemotherapeutics drugs with cytotoxic effects—doxorubicin and paclitaxel—are typically applied for patients with metastasized BC. Other treatments, including gemcitabine, cisplatin derivatives, 5-fluorouracil, or vinorelbine, are also used. On the other hand, combined treatments with chemotherapy drugs are considered a promising approach for enhancing BC therapy outcomes [147]. For instance, the estrogen antagonists—tamoxifen or fulvestrant—combined with the aromatase inhibitors—anastrozole, letrozole, and exemestane—are used in the hormone-dependent (ER+/PR+) BC cells [148]. Also, bevacizumab, a monoclonal antibody therapeutic, targets VEGFR, hindering the angiogenesis pathway [149-151]. Another monoclonal antibody, trastuzumab, could be used in patients overexpressing the HER-2 receptor, combined with therapeutic hormonal drugs such as the selective HER-2 pathway inhibitors lapatinib [148,150,151]. Moreover, various emerging drugs have shown the potential to overcome hormonal therapy resistance when combined with hormonal drugs [146]. These included the cyclin-dependent kinases 4 and 6 (CDK4/6) inhibitors such as abemaciclib, palbociclib, and ribociclib [152]—which impact cell cycle progression—and inhibitors of PI3K/AKT/mTOR pathway such as buparlisib, pictilisib, pilaralisib, and voxtalisib [153,154]. On the contrary, TNBC—the most aggressive cells with abolished biomarkers expression—the classical chemotherapeutic drugs, such as taxanes, anthracyclines, and platinum agents, remained the exclusive therapeutic option [149-151], and they are currently used with/without the monoclonal antibody against VEGF bevacizumab, [155]. Recently, new targeted drugs were introduced and still undergo clinical trials for optimizing BC therapeutic outcome, including poly ADP-ribose polymerase (PARP) inhibitors—olaparib, talazoparib, veliparib, niraparib, and rucaparib for those with mutated BRCA1/2—[156-158], the antibody drug conjugate Glembatumumab vedotin, the androgen receptor inhibitor bicalutamide, and the anti- PD-1 monoclonal antibody pembrolizumab.

  1. Illustrate a figure showing how PD-L1 expression regulates EMT in breast cancer for better understanding.

Response:

Thank you for the comment. The authors added the following figure.

Figure 2. PD-L1-mediated EMT stimulation. The diagram highlights the downstream signaling of EMT in cancer. Interferon-gamma, IFN-γ; IFN-γ receptor 1, IFNGR1; IFN-γ receptor 2, IFNGR2; epidermal growth factor, EGF; hepatocyte growth factor HGF; Janus kinase, JAK; signal transducer and activator of tran-scription3, STAT3; nuclear factor-kappa B, NF-Æ™B; phosphoinositide 3-kinase, PI3K; protein kinase B, AKT; mammalian target of rapamycin, mTOR; Zinc Finger E-Box Binding Homeobox 1/2, ZEB1/2; snail family transcriptional repressor 1, Snail1; extracellular-signal-regulated kinase, ERK; mitogen-activated protein kinase, MEK; B-Raf Ser-ine/threonine-protein, BRAF; rat sarcoma, Ras; programmed death-ligand 1, PD-L1; transforming growth factor-beta, TGF-β; mothers against decapentaplegic, smad; Epithelial-mesenchymal transition, EMT.

  1. Provide limitations and future perspective in conclusion.

Response

Thank you for the suggestion, and the authors added the following (line 614-622):

Various polyphenolic compounds have been used in BC clinical trials. These compounds have demonstrated promising anticancer approaches in patients with various grades of BC. These approaches include anti-inflammatory, pro-apoptosis, various anti-tumor biomarkers, anti-steroid hormone production effects. Also, these polyphenolics increased the expression of CEA, VEGF, and RDS. On the other side, few limited studies have proved the potential of these compounds to impact EMT-underlying tumor metastasis through modifying miRNA’s expression. Hence, further comprehensive investigations are suggested to highlight and focus on the potential of these dietary polyphenolics to reverse or inhibit the challenged EMT process.

  1. Avoid grammatical mistakes and typo errors.

Response

Thank you for the suggestion. The authors reviewed revised the manuscript and did some corrections with the assistance of grammar software.

Round 2

Reviewer 2 Report

Authors carried out the corrections according to the comments. The manuscript can be accepted for publication.

Author Response

Point by Point Response to Reviewer

Thank you for the suggestion. According to the reviewer’s suggestion, the authors revised the needed revisions and highlighted them in yellow in the revised submitted manuscript as follows.

  1. Polyphenols modulating effects of PD-L1/PD-1 checkpoint and EMT-mediated PD-L1 overexpression in breast cancer cells.

Revised

revised (Line 2 and 3)

(Polyphenols modulating effects of PD-L1/PD-1 checkpoint and EMT-mediated PD-L1 overexpression in breast cancer)

  1. When taking orally—since the mouth is the most common route of administration for small molecule drugs [199]—these polyphenols might face many obstacles before reaching their site of action. The challenges may include poor aqueous solubility, weak oral absorption, low bioavailability, rapid metabolism, or fast systemic elimination [198].

Revised (Line 398-402)

When taking orally—since the mouth is the most common route of administration for small molecule drugs and nutraceuticals  [199]—these polyphenols might face many obstacles before reaching their site of action. The challenges may include poor aqueous solubility, weak oral absorption, low bioavailability, or fast systemic elimination  [198].

  1. Clinical trials in BC patients evidenced the potential of the dietary polyphenolic compounds to increase apoptosis while decreasing various tumor biomarkers [201,202], steroid hormones production [203,204], the expression of the carcinoembryonic antigen (CEA) and VEGF [205], and radiation dermatitis severity score (RDS) [206], in addition to its anti-inflammatory effects [207]

Revised (Line 410-414)

 Clinical trials in BC patients evidenced the potential of the dietary polyphenolic compounds to increase apoptosis while decreasing various tumor biomarkers [201,202], including steroid hormones [203,204], carcinoembryonic antigen (CEA), VEGF [205], and radiation dermatitis severity score (RDS) [206], in addition to anti-inflammatory effects [207].

  1. Various polyphenolic compounds have been used in BC clinical trials. These compounds have demonstrated promising anticancer approaches in patients with various stages of BC. These approaches include anti-inflammatory, pro-apoptosis, various anti-tumor biomarkers, anti-steroid hormone production effects. Also, these polyphenolics increased the expression of CEA, VEGF, and RDS. On the other side, few limited studies have proved the potential of these compounds to impact EMT-underlying tumor metastasis through modifying miRNA’s expression. Hence, further comprehensive investigations are suggested to highlight and focus on the potential of these dietary polyphenolics to reverse or inhibit the challenged EMT process.

Revised (614-618)

Various polyphenolic compounds have been used in BC clinical trials. These compounds have demonstrated promising anticancer effects in patients with various stages of BC. These effects include anti-inflammatory, pro-apoptosis induction, and suppression of various tumor biomarkers such as CEA, VEGF, and RDS